# Ice-VII-like molecular structure of ambient water nanomeniscus

Dongha Shin[1], Jonggeun Hwang[1] & Wonho Jhe[1]

Structural transformations originating from diverse rearrangements of the hydrogen bonding in water create various phases. Although most phases have been well investigated down to the molecular level, the molecular structure of the nanomeniscus, a ubiquitous form of nanoscale water in nature, still remains unexplored. Here, we demonstrate that the water nanomeniscus exhibits the stable, ice-VII-like molecular structure in ambient condition. Surface-enhanced Raman spectroscopy on trace amounts of water, confined in inter-nanoparticle gaps, shows a narrowed tetrahedral peak at 3340 cm$^{-1}$ in the OH-stretching band as well as a lattice-vibrational mode at 230 cm$^{-1}$. In particular, the ice-VII-like characteristics are evidenced by the spectral independence with respect to temperature variations and differing surface types including the material, size and shape of nanoparticles. Our results provide un unambiguous identification of the molecular structure of nanoconfined water, which is useful for understanding the molecular aspects of water in various nanoscale, including biological, environments.

[1] Center for 0D Nanofluidics, Institute of Applied Physics, Department of Physics and Astronomy, Seoul National University, Seoul 08826, Republic of Korea. Correspondence and requests for materials should be addressed to W.J. (email: whjhe@snu.ac.kr)

There has been increasing interest in how water changes its structure and properties when confined at nanoscale[1], for example, in the low-dimensional confinement in graphene[2–4] or carbon nanotubes[5–8]. Unlike such artificially prepared nanostructures, water nanomenisci are naturally and prevalently formed by capillary condensation between nearby surfaces or by ambient drying in micro- and nano-particulates, such as in wet sand[9], sticky flour[10], swelling clay[11] and frictional material[12]. The mechanical characteristics of the nanomeniscus have been explored quite comprehensively;[12–14] for example, the surface-tension force induced by the capillary bridge of water exerts strong adhesion to the dry particulates, so that their mechanical stability can be established. However, the molecular structure of the ubiquitous water nanomeniscus itself has not been addressed despite the expectation that its unusual mechanical property[12] may be closely related to its structural information at the nanoscale, and, even though there have been extensive theoretical studies regarding hydration (interfacial) water, experimental clarification of its molecular structure in ambient conditions has been very challenging. In this article, by using the natural confinement system of dried silver nanoparticulate we have succeeded in demonstrating its molecular structure by a common analyzing technique of Raman spectroscopy, showing that the water nanomeniscus has the similar hydrogen-bonding character of ice-VII even in ambient conditions. The results imply that, for example, the chemical reaction taking place in the water nanomeniscus should behave very differently with respect to that in the conventional bulk water or ice due to the substantially weakened hydrogen bonding configuration of nanoconfined water molecules. Moreover, from a technical point of view, our results pave the new and efficient way of using the SERS on the trace amount of water in the extreme (nanoconfined) environment.

The unique phase of ice-VII has been investigated widely in geology and planetary science, because it serves as an indicator for the existence of water or ice in inaccessible frontiers, such as Earth's deep mantle and extra-terrestrial outer space[15–17]. This is because the temperature and pressure ranges in the phase fields of the bulk ice-VII indeed match those found in such extreme environments. Therefore, it has been commonly believed that ice-VII forms stably only in such extreme conditions[18,19]. Interestingly, our optical spectroscopic findings show that ice-VII-like molecular structure can also exist as a highly stable state of water even in normal terrestrial ambient conditions.

## Results

### Surface-enhanced Raman spectroscopy of water nanomeniscus.
To uncover the molecular structure of the water nanomeniscus, we used vibrational (Raman) spectroscopy, providing precise and unambiguous molecular information by observing the unique fingerprints of hydrogen bonding (HB) configurations[20–23]. In particular, surface-enhanced Raman spectroscopy (SERS) is a powerful platform that sensitively detects trace amounts of analytes present in the highly confined (nanoscale) space[24]. However, detection of the SERS signal of water still remains very challenging due to water's unusually low Raman scattering cross section, low adsorption ability on metal (i.e. no chemical bonding) and high vapour pressure. Therefore, SERS of water has been achieved so far only at the bulk scale using contaminated water in rather uncontrolled environment such as on the highly negative electrochemical electrode in the presence of halide ions, which unavoidably complicate unambiguous identification of the pure confinement effect on the molecular structure of water[25,26]. Our critical strategy to successful SERS of pure water lies in the ultra-high purification of the silver-nanoparticle solution without

the complication of adding any ions or applying electrochemical potential. We have achieved a dilution factor of over $10^6$, which is high enough to remove the impurities left in the solution after synthesis so that they cannot occupy the nanometric hot-spot region of SERS. This simple process of purification realizes an ideal nanoconfinement platform sufficient for structural investigation of the nanoconfined water. Specifically, we have repeatedly performed precipitation and dispersion of the solution to decrease the concentration of impurities below 1 nM to maximize the impregnation possibility of pure water in the nanogaps within the silver nanoparticulates when dried in ambient condition. Notice that during such a purification process, the aggregation phenomenon is easily observed (even with naked eyes) due to the removal of the surface citrate, which is used to stabilize the nanoparticle as a suspension state. In this regard, the reason why SERS of the nanoscale water has not been successful seems to be mostly due to insufficient dilution and purification in the typical synthetic and purifying processes of noble metal nanoparticles.

Figure 1a shows schematically the water nanomenisci formed in the nanoparticulate gaps when the ultra-purified aqueous nanoparticle solution is dried in ambient condition (at room temperature and pressure with ~35% relative humidity). The optical hot spot represents the effective electromagnetic field region where most of the SERS signal comes from (Fig. 1b). Therefore, if the liquid is confined within the nanoscale hot-spot region, SERS of trace amounts of the liquid can be realized. Figure 1c and d present gradual appearance of the SERS signal of water as purification progresses. The initial solution was primarily diluted three-fold by deionized water and allowed to dry before the SERS measurement. At the corresponding stage 1 in Fig. 1c, the hot spot is largely occupied by the impurities due to insufficient purification so that SERS of water is only slightly observable (curve 1, Fig. 1d). Spectral change from stage 1 to 5 shows that the SERS signal is more pronounced as the impurities (e.g. nitrate anion, sodium cation and citrate anion) are further removed with thirty-fold dilution in each step after stage 1 (i.e. the total dilution factor is $2.43 \times 10^6$). Note that, in general, there might exist some organic species even in multiply purified water, which may affect the SERS measurements and thus one has to carefully check their possible contributions. In our experiment, due to the highly confined nanometric space (hot spot), only the small ionic species such as sodium cation that has a comparable molecular size of water (~0.3 nm, molecular weight 18 g/mol) might compete with water molecules in penetrating into the hot spot. Even in such a situation where some small impurities are still left, however, we observe that the DDAA peak is not much changed in its position and bandwidth independently of the number of purification processes (refer to the spectra in the purification steps 2–4 in Fig. 1d), which indicates that the small impurities do not contribute significantly to our SERS results. Moreover, more bulk organic species, such as KCl, $BH_4^-$, glycerol, PVP and ascorbic acid, which also have been used in our experiments are even more likely to be excluded from the hot-spot region, as indicated by the almost invariant DDAA peak shape.

Remarkably, at stage 5, two sharp and anomalous peaks are evident and appear at 3340 and 3493 $cm^{-1}$ (curve 5, Fig. 1d), substantially different from the normal Raman of the bulk water obtained in ambient condition (top black curve, Fig. 1d) that exhibits three main OH-stretching bands; DDAA, DA and free-OH[25] (see caption in Fig. 1). The typical DDAA and DA bands are significantly blue-shifted by 94 and 73 $cm^{-1}$, and also highly bandwidth-narrowed to 56 $cm^{-1}$ (from 220 $cm^{-1}$) and 75 $cm^{-1}$ (from 166 $cm^{-1}$), respectively, where the blue- (red-) shaded area represents the DDAA (DA) peak. Moreover, the slightly lowered fraction of free-OH peak in the SERS signal (from 8.9% for bulk

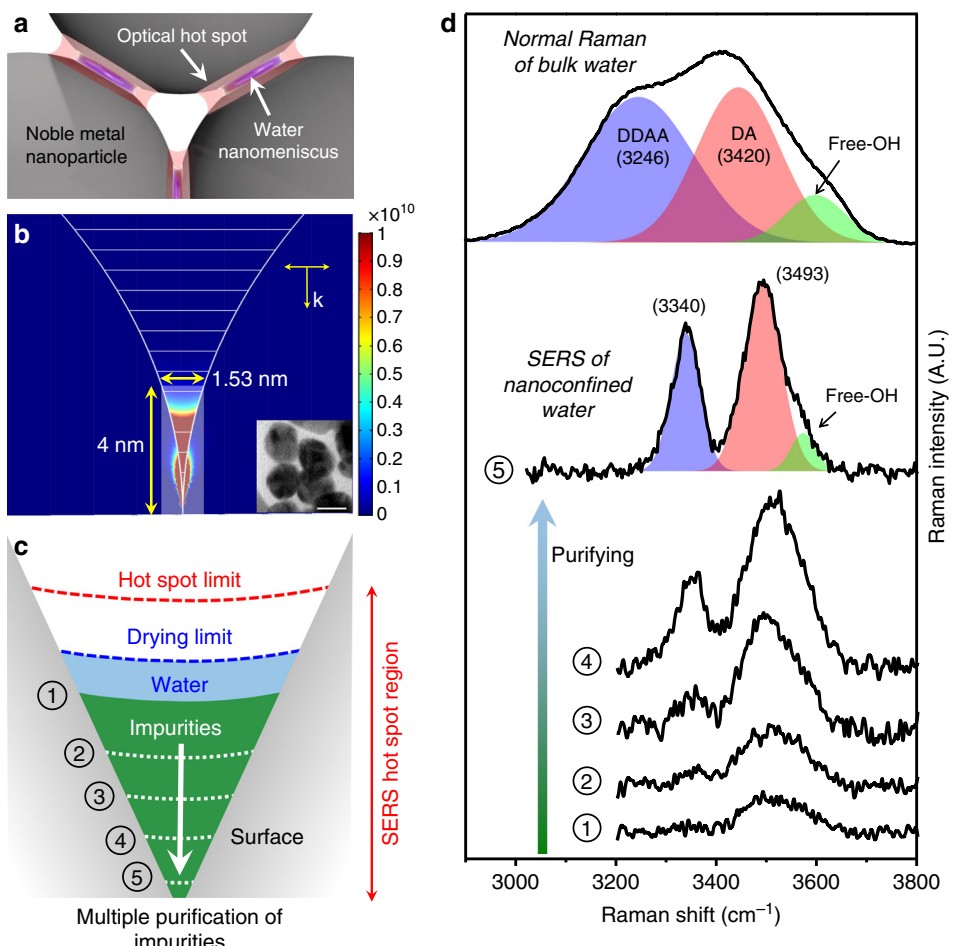

**Fig. 1** SERS for trace amounts of ultra-purified nanoconfined water. **a** Schematic showing the trapped water nanomeniscus in the optical hot-spot region after the highly purified silver nanoparticles (with the surface capped-citrate removed) are dried in ambient conditions. **b** Finite element method-based simulation of the electromagnetic fields shows that the field-enhanced region (or the SERS hot-spot) is highly localized at the crevice of silver dimer where nanoconfined water is formed. Here, the vertical scale represents the electromagnetic field enhancement ($= |$scattered field/incident field$|^4$). Inset shows the transmission electron micrograph (20 nm scale bar). **c** Schematic of the SERS hot-spot obtained by progressive purification. The hot-spot limit denotes the maximum optical region where most Raman signal comes from, while the drying limit indicates the maximum spatial region where nanoconfined water forms after drying. Stage 1 corresponds to the condition where most water is excluded from the hot spot due to the unremoved impurities remaining at the crevice, while stage 5 presents the situation where water occupies mostly the hot spot. **d** Spectra (1–5) show the increasingly prominent SERS signal of nanoconfined water as purification progresses, in contrast to the normal Raman of the bulk water (black curve, top) that shows three OH-stretching bands for the hydrogen bonding (HB) configuration; DDAA (D: HB donor, A: HB acceptor), DA and free-OH (for simplicity, negligible effects of DAA and DDA bands are omitted)

Raman to 7.8% for SERS, obtained by comparing the respective green-shaded area) suggests that water molecules are more hydrogen-bonded by tight confinement (DDAA or DA) and/or partly adsorbed on silver (DA) than bulk water[26,27]. Notice that the water nanomeniscus can be considered to consist of two components; the surface water layer (or strongly bound water) and the nanoconfined water (or weakly bound water), which are manifested by the DA and DDAA peaks, respectively. Notice also that 20 times stronger laser power (20 mW) was used to obtain the normal Raman spectrum of the bulk water. As a result, when we add pure water in the nanoparticulate system after stage 5, the SERS spectra in bulk water shown in Supplementary Fig. 1a (upper panel) is notably blue shifted from the normal Raman spectra of bulk water. This is due to the fact that a lower laser power (<1 mW) is used for SERS, so that the contribution from bulk water is negligible for the SERS spectra in bulk water and thus the probed region of nanoconfinement is only marginally increased.

**Ice-VII-like molecular structure of water nanomeniscus**. Interestingly, the abnormally narrowed OH-stretching bands of the SERS signal (DDAA and DA, Fig. 1d) strongly indicate the ice-like characteristics. In order to address its ice-likeness, we first obtain the bulk Raman spectrum of ice-Ih at 253 K (black curve, Fig. 2a) for comparison. As shown, we observe the intense peak near 230 cm$^{-1}$ in the low-frequency Raman bands, which represents the lattice-vibration band commonly observed in most ice phases, as well as the narrow high-frequency OH-stretching bands near 3140 cm$^{-1}$ (DDAA; the tetrahedral HB configuration of ice-Ih is shown in Fig. 2b). In contrast, the Raman signal of bulk liquid water obtained at 300 K (red curve, Fig. 2a) shows the usual OH-stretching bands that are blue-shifted with respect to the ice-Ih phase, but no sharp features are present both in the high- and low-frequency bands as expected. Now, in Fig. 2c, we present the full SERS spectra of the water nanomeniscus obtained at 300 K (black curve), whose high-frequency components are substantially narrowed (and blue-shifted) with respect to the bulk

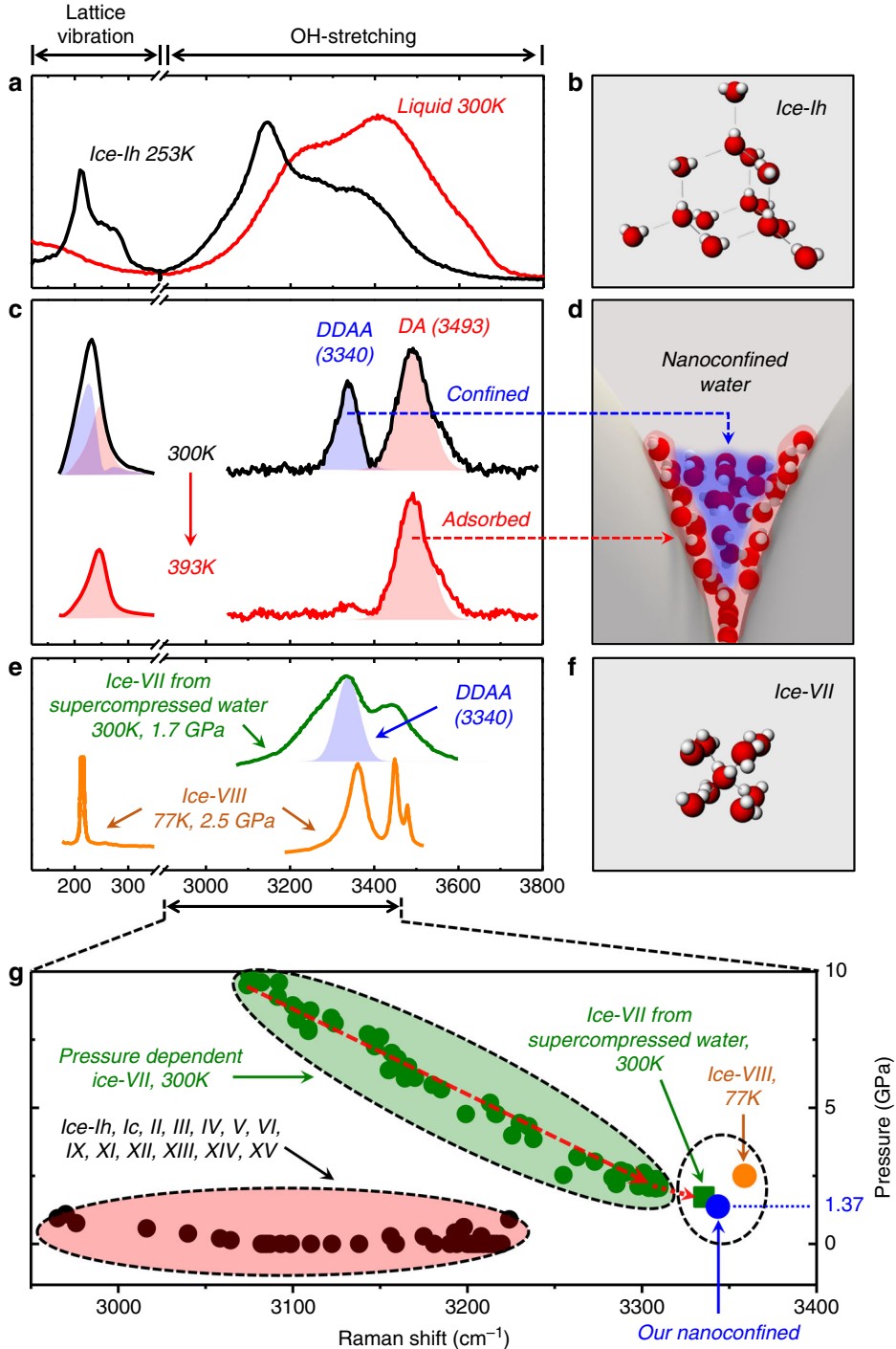

water, as already described in Fig. 1. Surprisingly, in particular, the intense sharp peak appears near 230 cm$^{-1}$ in the low-frequency bands of Fig. 2c, which demonstrates that the water nanomeniscus indeed possesses the ice-like characteristics.

For further characterization of the ice-like SERS spectra, we heat the nanoparticulate system and compare the black curve (before heating at 300 K) with the red curve (after heating at 393 K), given in Fig. 2c. Interestingly, in high-frequency region, we find that only the DA component (red-shaded area) remains after heating, whereas the DDAA component (blue-shaded area) disappears. Such selective decrease of the DDAA component supports that the DDAA peak originates from the nanoconfined water molecules (blue-dashed arrow, Fig. 2d), whereas the DA

peak is associated with the tightly bound water molecules on the silver surface (red-dashed arrow, Fig. 2d) that remain bound even after heating[26,27]. We also observe similar selective decrease of the component in the low-frequency region. Notice that the blue shades in Fig. 2c result from subtraction of the red curve from the black one. In summary, such correlated intensity-decrease, upon heating, between the two (low and high) frequency bands of the SERS spectra confirms that the water nanomeniscus exhibits the lattice-vibration characters of ice and the DDAA (DA) peak can be attributed to the nanoconfined (surface-adsorbed) water molecules.

Regarding the origin of the low-frequency bands near 200 cm$^{-1}$, there have been several simulation and experimental reports on

**Fig. 2** Ice-VII-like molecular structure of water nanomeniscus. **a** Normal Raman spectrum of bulk liquid water (red curve, at 300 K) and ice-Ih (black curve, 253 K). The room-temperature signal shows the broad OH-stretching bands in the high-frequency bands (also shown in Fig. 1d) and the broad features in the low-frequency bands. For ice-Ih, on the other hand, there exists the intense lattice-vibration band near 230 cm$^{-1}$ as well as the strong DDAA peak (~3145 cm$^1$). **b** The ordered structure of ice-Ih originates from the enhanced tetrahedral order of HB. **c** The full SERS spectra of the nanomeniscus in ambient condition (black curve, 300 K) as well as at 393 K (red curve) are presented. The SERS peak (3340 cm$^{-1}$, blue shade) at 300 K is significantly blue-shifted from ice-Ih but with the similar bandwidth (~50 cm$^{-1}$), which implies a distorted tetrahedral HB configuration while preserving the ice-like crystalline structure. We observe two heating-induced effects: correlated intensity-decrease of the DDAA component in the high- and low-frequency bands, as well as selective intensity-decrease of only the DDAA peak within each band. **d** The results of heating show that the nanoconfined water molecules contribute to the DDAA mode (blue shade) as indicated by the blue-dashed arrow, while the tightly surface-adsorbed water molecules produce the DA mode (red shade) (see the red-dashed arrow). **e** Raman spectra of ice-VII from supercompressed water (green curve) and ice-VIII (orange curve). Unlike the ice-VII-like confined water obtained at room temperature, ice-VIII is formed at low temperature. **f** The molecular structure of ice-VII is known as the densest phase of ice with the body-centered-cubic symmetry. **g** Thorough investigation of the available Raman bands of all ice phases (see Supplementary Fig. 3 for their references) shows the unique route where the SERS position of ice-VII (3340 cm$^{-1}$, ~1 GPa) can be located only by a slight extrapolation of the ice-VII from supercompressed water (3335.5 cm$^{-1}$, 1.72 GPa) at 300 K. Note that all the Raman spectra reported in the references were performed by normal Raman spectroscopy, not by SERS

the existence of translational bands even in the small water clusters[28–30], which have a very similar size compared to our nanomeniscus system. Moreover, the heating-induced correlated decrease between the low (lattice vibration) and high (OH stretching) frequency bands (Supplementary Fig. 2) strongly supports that the low-frequency bands should originate from the water character. In addition, since the lattice vibration bands (acoustic phonon modes) of silver metal nanoparticle exhibits in the range of 10–100 cm$^{-1}$, far below 200 cm$^{-1}$, they have nothing to do with the bands near 200 cm$^{-1}$ observed in our nanomeniscus[31–36]. All these results support that the low-frequency bands in our nanomeniscus can be attributed to the lattice vibration modes of water (ice), not to the silver metal.

Let us now focus on the DDAA contribution that represents the nanoconfined water. In order to identity the specific ice-like origin of the DDAA component, we have compiled all the reported Raman spectra of most ice phases in the OH-stretching region (Fig. 2g), obtained at various temperatures and pressures (see Supplementary Fig. 3 and its references). In the usual Raman studies of ice phases, the tetrahedral peak (DDAA) is generally used as a structural indicator because it represents the HB configuration very precisely. By close examination, we conclude that ice-VII may represent the molecular structure of our nanoconfined water based on the following two grounds: At first, for ice-Ih to ice-XV (except ice-X that has no OH-stretching band), the tetrahedral peaks are located only between 2950 and 3225 cm$^{-1}$ (black dots, Fig. 2g), more than 100 cm$^{-1}$ below the SERS peak (3340 cm$^{-1}$, blue dot); secondly, the peak position of ice-VII increases significantly in frequency with pressure drop, reaching maximum (3310 cm$^{-1}$) at 2.06 GPa (see green dots; the red-dashed line in green-shaded area is an eye-guide)[20], the lowest pressure achieved so far at 300 K (refer to the phase diagram of ice-VII, Supplementary Fig. 4). Notice that the highest frequency available for ice-VII is still 30 cm$^{-1}$ lower than the SERS of our water nanomeniscus, which is addressed in detail below.

Recently, a further increase of the DDAA frequency was reported at a lower pressure in the ice-VII that was directly crystallized from supercompressed water[37,38]. The authors used the dynamic diamond anvil cell to control the speed of pressurization on the water, which allows time-resolved analysis for the phase transformation pathways. They also found that the interfacial energy difference between the supercompressed water and ice-VII is smaller than that between the supercompressed water and ice-VI, manifesting the similarity of local order for the supercompressed water with respect to ice-VII, which is the body-centered-cubic (bcc) structure (Fig. 2f). Their ice-VII from supercompressed water exhibits Raman spectrum (green curve,

Fig. 2e), which has its peak position at 3335.5 cm$^{-1}$ (green square, Fig. 2g) even at 1.72 GPa, almost 30 cm$^{-1}$ lower than the one in the normal ice-VII (marked by a short red-dotted arrow). Remarkably, our nanomeniscus water shows the peak position at 3340 cm$^{-1}$, very similar to that of ice-VII from supercompressed water, which has a dense structure.

In addition, we would like to note that similar nanoconfinement-induced supercompression effect in ambient condition has been reported in other previous experiments and theory. Experimentally, the KI (potassium iodide) nanocrystal confined in carbon nanohorns (2 nm diameter) showed the super-high-pressure B2-phase structure (bcc), which occurs in bulk KI crystals above 1.9 GPa[38]. Moreover, the conducting sulphur phase was also obtained inside carbon nanotube, observable only at bulk scale above ~90 GPa[39]. Theoretically, molecular simulation on the pressure tensor of argon molecules confined in nanoslit showed the highly enhanced pressure near the slit surface[40,41]. In a similar way, the nanospace available in the purified silver nanoaggregates and the fast timescale of ambient-drying process in our experiment may render a supercompression-like effect, exerting high pressure on the nanomeniscus. Simple estimate also shows the pressure exerted is on the order of ~1 GPa (Supplementary Notes 1), consistent with the extrapolated pressure expected at the measured peak of 3340 cm$^{-1}$ (Fig. 2g).

Notice that ice-VIII[42] may have its DDAA-peak frequency similar to ice-VII (orange curve, Fig. 2e). However, the phase field of ice-VIII covers only the low-temperature range (i.e. 77 K at 2.5 GPa) unlike ice-VII. Moreover, since ice-XVI[43] and XVII[44] have too bulky structure of clathrate to exist in the nanometric hot spot and various bulk amorphous-ice phases[22] have much broader Raman bandwidths than those of the nanoconfined water, they cannot be the candidate either. Notice also that although infrared spectroscopy on the isolated water nanoclusters (formed at ~100 K)[45,46] showed the increase of the OH-stretching peak from 3200 to 3400 cm$^{-1}$ with the decrease of cluster size, our system is apparently different due to the confinement by hard surfaces, as evidenced by the crystalline bandwidth much narrower than those amorphous clusters (Supplementary Fig. 5).

**Generality of ice-VII-like molecular structure.** Figure 3a, b presents the temperature dependence of the SERS signal. As shown in Fig. 3a, the height of the DDAA peak (yellow-shaded area) decreases as temperature increases up to 393 K while the spectral widths are almost invariant, which justifies the assignment of the DDAA component to the nanoconfined water molecules (Fig. 2d) that can be 'evaporated' by thermal heating.

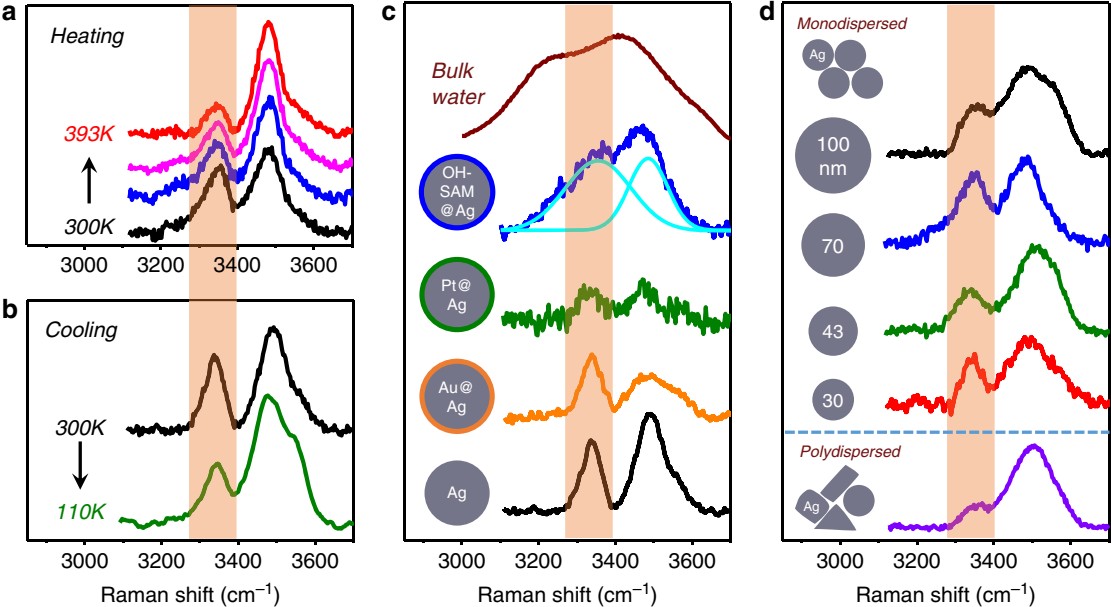

**Fig. 3** Generality of ice-VII-like molecular structure. **a** The DDAA and DA peaks are clearly observed during heating with their spectral widths almost invariant, which confirms the persistence of the ice-VII molecular structure for the water nanomeniscus even up to above the normal boiling temperature (393 K). Refer to text for the slight variations of the height of each peak. **b** The DDAA and DA peaks also show the qualitatively identical features at low temperature of 110 K. The combined results of **a** and **b** demonstrate the ice-VII-like molecular structure for the water nanomeniscus over a wide range of temperature. The noticeable change of the height and width of the low-temperature SERS signal is discussed in text. **c** Different kinds of metallic or nonmetallic surfaces coated on the silver nanoparticles are used to show the generality of the ice-VII-like characters for the water nanomeniscus that can be formed between various surfaces. Presented are the SERS signals for the gold-coated silver (orange curve), platinum-coated silver (green) and OH-SAM (2-mercaptoethanol)-coated silver (blue curve deconvoluted by two cyan curves), as well as the intrinsic silver nanoparticles (black) and the bulk Raman spectrum (brown curve at the top). **d** Normalized SERS signals demonstrate the similar ice-VII features for the water nanomeniscus, independently of its size, which is demonstrated by using several monodispersed as well as polydispersed nanoparticles (details of sample fabrication and characterization are presented in Supplementary Figs. 10 to 13)

On the other hand, Fig. 3b presents the results when temperature is decreased from 300 to 110 K, which still shows the very similar DDAA peak at ~110 K. For comparison, we observed an appreciable increase of the phonon-band frequency in the silicon substrate (semiconductor crystal) cooled at 110 K (Supplementary Fig. 6), which also confirms that the nearly invariant low-temperature DDAA spectra represent the water molecules behaving as the highly confined and stable phase of water. In short, therefore, the DDAA spectra of Fig. 3a, b clearly exhibit the ice-VII-like characteristics independently of temperature. We would like to emphasize, nonetheless, that although the SERS spectrum exhibits the same vibrational response of ice-VII, it does not necessarily mean that the structural properties, such as the molecular density distribution, are the same, which has to be further investigated.

Notice that, in Fig. 3a, the intensity of the DA peak increases slightly with temperature, which suggests that some of the nanoconfined molecules are thermally 'evaporated' and then re-adsorbed on the nanoparticle surfaces. Notice also that, in Fig. 3b, there are some spectral changes of the DA peak at 110 K; its maximum-peak position is slightly red-shifted, a small peak appears on its high-frequency shoulder (3550 cm$^{-1}$) and the overall DA intensity increases slightly in contrast to the DDAA one (see green curve, Fig. 3b). These low-temperature results of the DA peak may be associated with a phase transition to ice-VIII that exists only at low temperature and has the differing hydrogen configuration[42] (the ice-VIII spectrum is slightly different with respect to ice-VII, as shown in Fig. 2e). However, the above two propositions should be examined further in detail as a future work, because such unusual temperature-induced phase-change

behaviours seem to occur only in the highly confined (nano-confined) geometrical space, where not much is still known about phase transformation therein.

Figure 3c and d demonstrates the generality of the molecular structure of the water nanomeniscus, independent of the specific surface properties, by performing SERS for various nanoparticles having different kinds of surfaces. First, we fabricated the silver-gold core-shell nanoparticle (Au@Ag) to investigate the gold surface-confined water meniscus. For this purpose, instead of using the pure gold nanoparticles, we used the ultra-thin gold-coated silver particles that allows the silver-dominant plasmonic enhancement together with the gold-surface character (fabrication method and characterization is provided in Supplementary Figs. 7 and 8). The surface-adsorbed DA peak for the Au@Ag nanoparticle (orange curve, Fig. 3c) shows the spectral features somewhat broadened with respect to those for the pure silver nanoparticle (black curve, Fig. 3c), while, interestingly, the confined DDAA peak is very similar in both peak position and bandwidth to the silver case. Moreover, SERS of the platinum-coated silver nanoparticle, Pt@Ag, also shows the similar behaviours of DDAA (green curve, Fig. 3c), but with the smaller signal-to-noise due to the poor electromagnetic enhancement factor of platinum compared to gold or silver (see also Supplementary Figs. 7 and 8). Note that there might exist a strain on the synthesized core-shell nanoparticles so that the surface metal atoms will conform to the underlying lattice of the Ag shell[47]. However, our gold or platinum shell has more than 3 nm thickness, much beyond the 1–2 monolayer-thickness shell of atoms (refer to the EDS elemental line scan in Supplementary Figs. 7d and 7h), and the highly decreased SERS activity in Pt@Ag

structure also supports the existence of thicker shell of platinum, which contributes to the poor plasmonic optical activity (i.e. signal enhancement in SERS) compared to gold or silver. Therefore, the Au@Ag or Pt@Ag nanoparticle used in our study exhibits the pure gold- or platinum-like surface characteristics, which is manifested by the different NC peak shift in Supplementary Fig. 8.

For another generality test of the molecular structure of the water nanomeniscus, we used a common nonmetallic surface, a silica surface that is fabricated by incorporating the hydrophilic OH group on the silver nanoparticle (OH-SAM@Ag), which is based on the well-established thiol-based self-assembled monolayer (SAM) (2-mercaptoethanol). The SERS spectrum in Fig. 3c (blue curve) shows similar peak position (see two deconvoluted cyan curves) with respect to the DDAA peak of the pure silver case (black curve) while the bandwidth is much broadened. Such a broadening may originate from the reduced electromagnetic hot-spot region where the nanomeniscus forms, resulting from the existence of the finite-thickness dielectric layer of alkyl chain in OH-SAM. This situation can be understood in a similar way to stage 1 in Fig. 1c, d, where the water is excluded from the hot-spot region due to the unremoved impurities remaining at the crevices (hot spot). This observation is also supported by the low signal-to-noise (S/N) ratio of the water meniscus peak in the blue curve (Fig. 3c), as well as the absence of the meniscus peak in the OH-SAM@Ag nanoparticle when the much longer alkyl chain is attached (Supplementary Fig. 9).

Figure 3d presents the general size independency of the nanomeniscus from experiments using various monodispersed as well as polydispersed nanoparticles, which shows again very similar spectral features of DDAA for all the samples (notice that the washing and drying conditions are all identical and the Raman intensities are normalized for comparison). More experimental details on the fabrication methods and characterization results as well as the corresponding SERS spectra including the low-frequency spectra are presented in Supplementary Information (see Supplementary Fig. 2 and Figs. 10 to 13). We emphasize that the results shown in Figs. 3c and 3d verify that the water nanomeniscus is formed universally regardless of the surface types of the nanoparticle in ambient condition and the DDAA peak of the nanomeniscus exhibits generally the ice-VII-like molecular characteristics in all circumstances (Supplementary Figs. 14 and 15).

## Discussion

Finally, from a different perspective, we address the intriguing question as to why the nanoconfined water has similar structure to specifically ice-VII instead of other phases available within the general water phase-diagram. Whereas most ice exists in low temperature condition, only VI, VII, X and ultra-high-pressure phase are stable at room temperature. However, X and ultra-high-pressure phase have the ionized or metallic super-structures, respectively, in which the generally accepted characteristics of individual water molecules seem to vanish. Therefore, although ice-VI or ice-VII can be a plausible candidate at room temperature, ice-VII has a much denser structure (Fig. 2f) than ice-VI, and therefore, can be attributed to the nanomeniscus of water that forms in the interfacial nanogaps at ambient temperature. Our findings may help to explore the unusual behaviours of water under nanoconfinement, as well as provide a better understanding of various chemical and biochemical reactions in nanoscale water. Notice that although we have performed a separate experiment of isotopically diluted HDO in $D_2O$, we could not stably observe the OH stretching spectrum, which is worthy of further systematic investigation.

## Methods

**Synthesis and purification of conventional silver nanoparticles**. We used the citrate-reduction method[48]. Ninety milligrams of silver nitrate (Aldrich, 97%) was dissolved in 500 mL of distilled water. When boiling occurs, 10 mL of 1% sodium citrate (Aldrich, 98%) solution was added and left for 20 min. Average particle size (~35 nm) was measured by transmission electron microscopy. Silver sol contained the unreacted impurities of millimolar concentration, such as the nitrate anion, sodium cation and citrate anion. In each stage of purification, the diluted solution (1 mL) was precipitated by centrifugation (~15,000 rpm), discarded (990 μL) and redispersed with pure distilled water (10 + 990 μL). For stage 1–5, we obtained the concentration of impurities as 0.33 mM, 11.11 μM, 0.37 μM, 12.34 nM and 0.41 nM, respectively.

**Numerical simulation**. We used the Finite Element Method [COMSOL Multiphysics (Stockholm, Sweden), Wave Optics Module] to calculate the local electromagnetic fields (at 532-nm wavelength) between the adjacent spherical silver nanoparticles (two spheres of 35-nm diameter and 0.6-nm interparticle gap). The refractive index value provided by Johnson and Christy[49] was used for silver at the excitation wavelength, and the perfectly matched layers were used to absorb the outgoing waves. The incident wave light was linearly polarized parallel to the interparticle axis while propagating perpendicular to the same axis.

**Raman measurement**. After loading of 10-μL purified sol on the silicon wafer substrate by a micropipette, the sample was sufficiently dried in ambient condition (300 K and 35% relative humidity). Then, Raman measurements were conducted using the Witec Alpha 300 system (532-nm laser light with ~2-μm spot size, 90-s integration time, grating of 600 g/mm for instantaneous measurement of wide spectral range). Relatively low laser power (<1 mW) was used for SERS to avoid the local heating that might induce structural deformation of silver by surface plasmon resonances, as well as to avoid the photo-related chemical reaction such as photobleaching. Higher laser power (20 mW) was used only to obtain the normal Raman of the bulk water near 300 K. Notice that when measuring the SERS of the water nanomeniscus with low laser power (< 1 mW), especially when the bulk water was added in the nanoparticulate system (Supplementary Fig. 1), the contribution of the bulk-phase water in the Raman signal was negligible due to the intrinsically low Raman cross section of the bulk water. Temperature-dependent Raman measurements were made using the hot and cold stage (INSTEC HCS302, from −160 to 400 °C) while the low-temperature condition was provided by the LN2-P8UD10 liquid nitrogen cooling system.

**Nanoparticle characterization**. The UV-vis absorbance spectra of the various nanoparticles were recorded using a SILVER-Nova spectrometer (StellarNet). Scanning Transmission Electron Microscopy images with EDS analysis were taken using a FEI Tecnai F20 microscope operated at 200 kV for Au@Ag nanoparticle and JEOL JEM-2100F microscope operated at 200 kV for Pt@Ag one. Field-emission scanning electron microscopy image was taken using the AURIGA model (Carl Zeiss).

## Data availability

The data sets generated and/or analysed during the current study are available from the corresponding author on reasonable request.

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

## Acknowledgements

The authors acknowledge Hansol Song for performing FEM simulation, Prof. Sangmin An for helpful discussions and Brendan Deveney for critical reading of the manuscript. This work was supported by the National Research Foundation of Korea (NRF) grant funded by the Korea government (MSIP) (No. 2016R1A3B1908660) and Basic Science Research Program through the National Research Foundation of Korea (NRF) funded by the Ministry of Education, Science and Technology (2017R1A6A3A11031278).

## Author contributions

D.S. and W.J. designed and led the project and wrote the manuscript. D.S. and J.H. carried experiments, and D.S. and W.J. performed data analysis. All authors discussed and commented on the manuscript.

## Additional information

**Competing interests:** The authors declare no competing interests.

