## [Peer Review File · Nature Communications]

Reviewers' comments:

Reviewer #1 (Remarks to the Author):

The manuscript by Shin et al. titled "Ultrasensitive Identification of Molecular Structure of Ambient Water Nanomeniscus" presents new measurements of surface-enhanced Raman spectroscopy on nanoconfined water between Ag nanoparticles. The manuscript makes a compelling case that the local molecular structure of nanoconfined water between Ag nanoparticles after drying resembles that of ice VII, which I find significant and could merit publication in Nature Communications. The only finding I am aware of that is somewhat related to this is Corsetti et al. [Phys. Rev. Lett. 116, 086901 (2016)] that found a disordered triangular bilayer of nanoconfined ice with high configurational entropy and high density, but this study was for a spatially extended 2D system and the findings only supported by simulations.

Nevertheless, I have some concerns that should be addressed before I would support the publication of the manuscript in Nature Communications.

- One could discuss whether it is relevant to call such nanoconfined water ice like if it only has a few (at most 5) waters in width at the measured hot spot. Thus, the sharp peak at ~ 200 cm^{-1} associated with lattice vibrations must be mediated by the lattice vibrations of the Ag nanoparticles, since there is no "water lattice" to talk about in the crevice with such limited translational order. The authors are quite careful to call it molecular structure and not crystal structure of ice-VII, but this could be further stressed. I would specifically refrain from talking about "phase transitions" since the confinement is so severe that translational order is almost completely lacking. I would like the authors to clarify how lattice vibrations can occur in nanoconfined water when translation order, which is present in ice VII, cannot be present in the water nanomeniscus up to more than at most 3-4 unit cells of ice VII.

- The scientific arguments used to support the claim that the local molecular structure in the nanomeniscus is similar to ice VII is based on the fact that vibrational resonances probed by SERS, especially the one at 3340 cm^{-1} associated with DDA H-bonds, have the same frequencies as the vibrational resonances probed by normal (bulk-sensitive) Raman spectroscopy of ice VII shown in Fig. 2e-g. However, the SERS spectra in bulk water shown in Fig. S1a (upper) is notably blue shifted from the normal Raman spectra of bulk water. I assume this is due to the fact that lower laser power is used for SERS, so that the contribution from bulk water is negligible for the SERS spectra in bulk water and the probed region of nanoconfinement is only marginally increased. However, this does not become clear

until reading about the Raman measurement in the Methods section on p. 11-12 and should be stated more clearly in the main text of the manuscript.

MINOR COMMENTS

Additionally, I have a few minor comments that also should be addressed in the revised manuscript:

- How is the impurity concentration below 1 nM confirmed? High-quality (type 1) water usually contains less than 50 $\mu\text{m}/\text{l}$ organics (see NIH report on "Laboratory Water - Its Importance and Application", March 2013), which corresponds to roughly 10-100 nM depending on molecular weight of the organics. Although the dilution process evidently improves the water spectra presented in Fig. 1d, I doubt that the author's can ensure that the concentration of impurities is as low as 0.41 nM.

- The dilution factor from stage 1 to stage 5 with thirty-fold dilution is $8E5$, not over $1E6$.

- There is a small high-energy shoulder in the stage 5-purified SERS spectra of water, which could correspond to a very low fraction of free OH bonds. What fraction of free OH bonds are fitted if a 3-peak model fit is used for nanoconfined as well as bulk water?

- Clarify in figure caption of Fig. 2g that the ice spectra are taken from literature, by e.g. changing "all ice phases shows" on line 424 to "all ice phases (see Fig. S2 for their references) shows". Furthermore, are all spectra performed with normal (bulk-sensitive) Raman spectroscopy or SERS?

- Please indicate the estimated internal pressure of ~ 1.37 GPa of your blue data point measured at 300 K in Fig. 2g.

- Please change "supercompressed-ice-VII" to "ice VII from supercompressed water" at 300 K, since it is not (according to ref. 16 of the SI) the ice VII phase that is supercompressed but the liquid water phase from which it is transformed.

- How does the peak at $\sim 200\text{ cm}^{-1}$ presented in Fig. 2c look like at drying stages 1-4 and for the various nanoparticle systems presented in Fig. 3c-d? Since I suspect the lattice vibrations would be mediated by the lattice vibrations in the nanoparticles, it would be highly interesting to see whether the peak changes for different systems. Furthermore, could the peak be assigned to nanoparticles or impurities solely, or does it only occur for nanoconfined water in between nanoparticles?

- There will be a strain on the synthesized core-shell nanoparticles so that the surface metal atoms will conform to the underlying lattice of the Ag shell. Thus, although it alters the chemical activity of the surface, it does not correspond to a pure nanoparticle of the shell element, see e.g. enhanced catalytic activity of Pt@Ru in Alayoglu et al., [Nature Mater. 7, 333 (2008)].

LANGUAGE

The manuscript conforms to the Nature Communications guidelines, but the language needs improvement. See specific suggestions below:

- ice phases are usually written with dash, e.g. "ice VII"
- add space between temperature and K, e.g. "300 K"
- p. 2, line 25: "carbon nanotube" \rightarrow "carbon nanotubes"
- p. 2, line 27: "the wet sand" \rightarrow "wet sand"
- p. 3, line 48: "anlaytes" \rightarrow "analytes"
- p. 4, line 79: "is slightly observable" \rightarrow "is only slightly observable"
- p. 5, line 87: "Obviously, the typical ..." \rightarrow "The typical ..."
- p. 7, line 154: ".," \rightarrow "."
- p. 8, line 193: "(Au@Au)" \rightarrow "(Au@Ag)"
- p. 10, line 229: "similar structure to the specific ice-VII" \rightarrow "similar structure to specifically ice VII"
- p. 10, line 231: "can cover the stability fields at room temperature" \rightarrow "are stable at room temperature"
- p. 22, line 239: "in the nanoscale water" \rightarrow "in nanoscale water"
- p. 20, line 403: "of the bulk liquid water" \rightarrow "of bulk liquid water"

- p. 20, line 405: “the smooth features” → “the broad features”
- p. 20, line 412: “implies the highly distorted” → “implies a distorted”
- p. 20, line 421: “at extremely low temperature” → “at low temperature”
- p. 20, line 426: “supercompressed-ice-VII” → “superexpanded ice VII” or “supercompressed water”
- p. 21, line 429: “ice-VII structure” → “ice-VII molecular structure”
- p. 21, line 441: “gold coated silver” and “platinum coated silver” → “gold-coated silver” and “platinum-coated silver”
- p. 21, line 442: “OH-SAM (2-mercaptoethanol) coated silver” → “OH-SAM (2-mercaptoethanol)-coated silver”
- p. 21, line 446: “its various size” → “its size”

Reviewer #3 (Remarks to the Author):

Dear Editor

I have read the manuscript titled “Ultrasensitive Identification of Molecular Structure of Ambient Water Nanomeniscus” by Shin and coworkers.

In their work, the authors report on surface-enhanced Raman spectroscopy study of water confined in inter-nanoparticle gaps. Based on the observed Raman spectra, the authors make conclusion that the water nanomeniscus has the stable molecular structure of ice-VII in ambient condition. I think this conclusion is absolutely wrong. The observed peaks in Raman spectrum (Fig. 1d) originates from the interfacial water, or reflects the PREWETTING transition, i.e. formation of a liquid layer at a surface. This phenomenon has been intensively studied for water in a slit pore of width of 24 Å (J. Phys. Chem. C 2007, v. 111, 15716), which is about the same size of inter-nanoparticle gaps. The peculiarity of water is the formation of water bilayer (water monolayer) upon prewetting transition. This water bilayer (monolayer layer), together with interfacial water is known as “bound water”, which show strong positional and orientational ordering (ice-like). Thus this manuscript is not deserving publishing in Nat. Comm.

In what follows, I will give few reasons to support my choice.

1- According to the water phase diagram, Ice VII is formed from liquid water above 3GPa (very high pressure). However, without applying pressure in the experiments, the claimed conclusion that the water nanomeniscus has the stable molecular structure of ice-VII is unconvincing. Moreover, the highest frequency available for ice-VII is 30 cm^{-1} lower than the obtained SERS of water nanomeniscus (Fig. 2g) can further support that the molecular structure of water nanomeniscus is different with that of ice-VII, although the authors tried to explain the difference.

2- Ice-VII has a density of about 1.65 g/cm^3 , which is much higher than the density of bulk water. As we known, the water layers next to the hydrophilic surface is very dense, and maybe that's why observed SERS of water nanomeniscus is like that of ice-VII.

3-After heating at 393K, the DA component remains, which indicates that the bound water molecules on the silver surface is ice-like (Fig. 2c). The disappearance of the DDAA component maybe related to the second layering transition or the disappearance of the second water layer near surface.

Reviewer #4 (Remarks to the Author):

In this paper Shin et al use Raman spectroscopy to identify the dominating form of molecular organisations in water nano-menisci between metal nanoparticles at room temperature. They find unambiguously the water to adopt the molecular structure of ice-VII.

The results are in my view remarkable because ice-VII is normally expected only for water under extreme conditions of high pressure or low temperature. The existence of water in ice-VII form at room temperature opens up questions regarding our understanding of water behaviour at nano-interfaces. This has potentially some broad implications given the ubiquity of water nano-menisci in nature and technology.

However, while I find the results in principle exciting, I have some serious concerns with the paper in its current form.

First, the paper almost exclusively focus on control experiments and misses the in my view the most important part: a thorough discussion about the implications of the findings (beyond the usual sweeping generalities). Do the findings have implications for the dynamics of water molecules in nano-menisci, and hence any chemical reaction that could take place? Are the results expected to stand in the presence of solutes? Do the findings have implications for the current use of Raman spectroscopy to identify water in extreme environments? Currently the paper is very dry and somehow very defensive, constantly trying to argue against possible objections. This makes the read off putting for non-Raman specialists (broad readership).

Second, I am not convinced about the generality of the findings as claimed in the abstract. It is true that the authors make every effort to probe interfaces beyond noble metals, but even the use of SAM-coated Ag NPs is in my view not enough to assert that water nano-menisci between the grains of a drying sand-castle are indeed ice-VII. Besides, the authors go to great length to purify their water beyond anything expected for a general/natural system.

Finally and in the same vein, perhaps I am missing something, but the NPs seem citrate-coated to me given the protocol used to create them. I am not specialist on Ag NPs, but Au NPs will spontaneously decompose without a protective layer of citrate or a SAM. To me this raises serious questions about the picture given in Figs 1-2: where is the citrate? And what is actually being measured in terms of adsorbed water? I don't necessarily think that this invalidates the findings/conclusion, but it at least deserves a proper discussion, especially since the authors contrast 'Ag NPs' with 'SAM-Ag NPs', the later exhibiting a signal considerably poorer.

Reviewer #5 (Remarks to the Author):

This is a very interesting paper that should be published. The use of SERS in conjunction with water capillaries is creative and reveals a rather unique spectral response of confined water. I have a few comments the authors may wish to consider.

For the sake of scientific rigor, I would suggest the authors to tone down the claim that the water is ice; it has the same vibrational response of ice, but that does not mean that the structural properties, like $g(r)$, are the same. It would not hurt, and improve the paper, I submit.

The authors should do at least one set of experiments with isotopically diluted H₂O/D₂O mixtures. It is well-known (see e.g. papers from Skinner, Bakker, and others) that the OH stretch vibration is

delocalized, especially in ice. This excitonic behaviour depends on the size of the water cluster under study. Hence, the most appropriate comparison would be to study the O-H stretch spectrum of isotopically diluted HDO in D₂O. The point is that it is not clear how many water molecules are in the meniscus relative to the delocalization length of the OH stretch.

The authors are advised to explore the role of the meniscus on the spectra: there will be a substantial LaPlace pressure, that may contribute to the high-pressure-type of behavior observed. A good starting point may be the book 'Intermolecular Forces' by Israelachvili

Reviewers' comments:

Reviewer #1 (Remarks to the Author):

The manuscript by Shin et al. titled "Ultrasensitive Identification of Molecular Structure of Ambient Water Nanomeniscus" presents new measurements of surface-enhanced Raman spectroscopy on nanoconfined water between Ag nanoparticles. The manuscript makes a compelling case that the local molecular structure of nanoconfined water between Ag nanoparticles after drying resembles that of ice VII, which I find significant and could merit publication in Nature Communications. The only finding I am aware of that is somewhat related to this is Corsetti et al. [Phys. Rev. Lett. 116, 086901 (2016)] that found a disordered triangular bilayer of nanoconfined ice with high configurational entropy and high density, but this study was for a spatially extended 2D system and the findings only supported by simulations.

Nevertheless, I have some concerns that should be addressed before I would support the publication of the manuscript in Nature Communications.

Question 1-1.

One could discuss whether it is relevant to call such nanoconfined water ice like if it only has a few (at most 5) waters in width at the measured hot spot. Thus, the sharp peak at ~ 200 cm⁻¹ associated with lattice vibrations must be mediated by the lattice vibrations of the Ag nanoparticles, since there is no "water lattice" to talk about in the crevice with such limited translational order. The authors are quite careful to call it molecular structure and not crystal structure of ice-VII, but this could be further stressed. I would specifically refrain from talking about "phase transitions" since the confinement is so severe that translational order is almost completely lacking. I would like the authors to clarify how lattice vibrations can occur in nanoconfined water when translation order, which is present in ice VII, cannot be present in the water nanomeniscus up to more than at most 3-4 unit cells of ice VII.

Response 1-1.

Thank you for valuable comments.

Firstly, regarding the possibility of the lattice vibration modes in the highly confined space, we would like to cite several related studies previously made on the small water clusters, which has a very similar size with respect to our nanomeniscus system.

1. Bosma *et al.* reported in their simulation work that there exist the low frequency Raman spectra for water clusters as a function of size (from N=2 to 50), which clearly shows the bands near 200 cm⁻¹. (Fig. 2 in *J. Chem. Phys.* 98, 4413 (1993))
2. Knochenmuss *et al.* also presented translational modes near 200 cm⁻¹ in N=5 and N=8 water clusters. (p. 5242 in *J. Chem. Phys.* 96, 5233 (1992))
3. More recently, Liu *et al.* reported in their experimental work that translational bands near 200 cm⁻¹ are observed in the FTIR jet spectra of water clusters. (Fig. 3 in *Phys. Chem. Chem. Phys.* 6, 3315 (2004))

In summary, all these reports suggest and demonstrate the existence of translational modes even in small water clusters.

Secondly, in our SERS spectrum we observed strong correlation between the low-frequency bands and the OH stretching bands. As already mentioned in the main text, heating experiment in Fig. 2c (in main text) clearly shows the selective decrease of DDAA peak (3340 cm^{-1}) in the OH stretching bands. At the same time, we also observed the selective decrease in the low-frequency bands (see the disappearance of the blue shaded area near 200 cm^{-1}). Moreover, we have performed additional heating experiments for both ultra-purified and less purified nanoparticle solution, and found that in both cases, there exists a strong correlation in peak intensity as well as peak width between the low-frequency bands and the OH stretching bands, which is shown in the following Fig. S2.

Figure S2. Correlated decrease of the peak-area ratio between the low (A/B) and high (C/D) frequency region is observed during heating experiment (120 °C for 20 min.). **a**, The results are obtained in a separate experiment and are similar to those presented in Fig. 2c in the main text for ultra-purified nanoparticle solution. **b**, The data show the heat-induced change of SERS spectrum of the nanomeniscus produced from less purified nanoparticle solution, which is characterized by the broadened OH stretching bands as is also shown in Fig. 1d in the main text.

Both Fig. S2a and 2b show that there is the heating-induced correlated decrease of the peak-area ratio between the low (A/B) and high (C/D) frequency component. Notice that much as the bandwidths in the OH stretching region in Fig. S2b are broader than those in Fig. S2a (compare C and D in Fig. S2a with those in Fig. S2b), there also exists a broadening of the bandwidths in the low-frequency region (compare A and B in Fig. S2a with those in Fig. S2b). Therefore, since we know that the high frequency bands (OH stretching bands) originate from the water molecules, the heating-induced correlated decrease observed between the low and high frequency region supports that the low-frequency peak also comes from the water character, not from the lattice vibration of Ag nanoparticle.

Thirdly, in many reports on the lattice vibration modes (acoustic phonon modes) of silver metal (even in silver nanoparticle), Raman spectrum corresponding to the lattice vibration modes exhibits in the range of $10\text{ cm}^{-1} \sim 100\text{ cm}^{-1}$, far below 200 cm^{-1} (see the references below). While some ionic silver compounds such as AgO and AgCl show the modes near 200 cm^{-1} in the normal Raman measurement, they have nothing to do with the OH stretching bands of water.

A. Nelet *et al.* Appl. Surf. Sci. **226**, 209, (2004)

A. Courty *et al.* J. Chem. Phys. **116**, 8074, (2002)

G. Bachelier *et al.* Phys. Rev. B **69**, 205408, (2004)

B. Palpant *et al.* Phys. Rev. B **60**, 17107, (1999)

I. Martina *et al.* E-Preservation Science: Morana RTD **9**, 1-8, (2012)

M. Fujii *et al.* Phys. Rev. B **44**, 624-3, (1991)

In summary, with the supportive results and grounds listed above, we have added supplementary discussions on the origin of lattice vibration bands of water nanomeniscus in the revised manuscript as follows. We also added Fig. S2 in Supplementary Information.

Revisions made 1-1.

In page 7, we added the following paragraph:

“Regarding the origin of the low-frequency bands near 200 cm^{-1} , there have been several simulational and experimental reports on the existence of translational bands even in the small water clusters²⁸⁻³⁰, which has a very similar size compared to our nanomeniscus system. Moreover, the heating-induced correlated decrease between the low (lattice vibration) and high (OH stretching) frequency bands (Fig. S2) strongly supports that the low-frequency bands should originate from the water character. In addition, since the lattice vibration bands (acoustic phonon modes) of silver metal nanoparticle exhibits in the range of $10\text{ cm}^{-1} \sim 100\text{ cm}^{-1}$, far below 200 cm^{-1} , they have nothing to do with the bands near 200 cm^{-1} observed in our nanomeniscus.³¹⁻³⁶ All these results support that the low-frequency bands in our nanomeniscus can be attributed to the lattice vibration modes of water (ice), not to the silver metal.”

Question 1-2.

The scientific arguments used to support the claim that the local molecular structure in the nanomeniscus is similar to ice VII is based on the fact that vibrational resonances probed by

SERS, especially the one at 3340 cm⁻¹ associated with DDAA H-bonds, have the same frequencies as the vibrational resonances probed by normal (bulk-sensitive) Raman spectroscopy of ice VII shown in Fig. 2e-g. However, the SERS spectra in bulk water shown in Fig. S1a (upper) is notably blue shifted from the normal Raman spectra of bulk water. I assume this is due to the fact that lower laser power is used for SERS, so that the contribution from bulk water is negligible for the SERS spectra in bulk water and the probed region of nanoconfinement is only marginally increased. However, this does not become clear until reading about the Raman measurement in the Methods section on p. 11-12 and should be stated more clearly in the main text of the manuscript.

Response 1-2.

Thank you for valuable comments.

We totally agree with the reviewer's opinion that the blue shift observed in the SERS of bulk water in Fig. S1 is due to negligible contribution from bulk water. According to the reviewer's suggestion, we have modified the main text to describe more clearly as follows.

Revisions made 1-2.

In page 5, we added the following sentences:

“Notice also that 20 times stronger laser power (20 mW) was used to obtain the normal Raman spectrum of the bulk water. As a result, when we add pure water in the nanoparticulate system after stage 5, the SERS spectra in bulk water shown in Fig. S1a (upper panel) is notably blue shifted from the normal Raman spectra of bulk water. This is due to the fact that a lower laser power (<1 mW) is used for SERS, so that the contribution from bulk water is negligible for the SERS spectra in bulk water and thus the probed region of nanoconfinement is only marginally increased.”

MINOR COMMENTS

Comment 1-3.

Additionally, I have a few minor comments that also should be addressed in the revised manuscript:

How is the impurity concentration below 1 nM confirmed? High-quality (type 1) water usually contains less than 50 µm/l organics (see NIH report on “Laboratory Water - Its Importance and Application”, March 2013), which corresponds to roughly 10-100 nM depending on molecular weight of the organics. Although the dilution process evidently improves the water spectra presented in Fig. 1d, I doubt that the author's can ensure that the concentration of impurities is as low as 0.41 nM.

Response 1-3.

The impurity concentration we mentioned indicates the ones left from the precursor chemicals of sodium citrate (Na⁺, C₆H₅O₇⁻) and silver nitrate (Ag⁺, NO₃⁻) used for nanoparticle synthesis. What is important in our success of realization of nanomeniscus SERS lies in enhancing the impregnation ability of water molecules into the highly confined nanometric space (hot spot). Due to this ultra-confined space for SERS, only the small ionic species such as sodium cation

that has comparable molecular size of water (~0.3 nm, molecular weight 18 g/mol) might penetrate competitively into the hot spot. Even in such a situation where some small impurities are still left, we observe that the DDAA peak is not much changed in its position and bandwidth (refer to the spectra in the purification step 2 to 4 in Fig. 1d in the main text). This indicates that the small ionic species do not contribute much to our SERS results. On the other hand, as the referee mentioned, more bulk organic species such as the citrate anion has even less capability to reside in the hot spot: for example, if 50 µg/l of organic compound (appearing in the NIH report) has 10 nM concentration, its molecular weight should be 5000 g/mol (100 nM corresponds to 500 g/mol), which is extremely large and thus likely to be excluded from the hot spot, without competing with water molecules. Even in such a case of various kinds of large impurities possibly present in the solution, such as KCl, BH⁴⁻, glycerol, PVP and ascorbic acid, we still observe almost invariant DDAA peak shape (see the notes in Section 11 and Fig. S14 in Supplementary Information). Therefore, our results show that the impurities do not interfere with the spectral shape of SERS spectrum of water nanomeniscus at least as far as water is concerned for SERS measurements. Nonetheless, we would like to carefully describe the possible impurity issue in the present work because as the reviewer commented, it is quite important in many analytical SERS studies.

Revisions made 1-3.

In page 4, complying with the reviewer's comment, we added the following sentences to notify the presence of organic species even in the multiply purified water environment.

“Note that, in general, there might exist some organic species even in multiply purified water, which may affect the SERS measurements and thus one has to carefully check their possible contributions. In our experiment, due to the highly confined nanometric space (hot spot), only the small ionic species such as sodium cation that has a comparable molecular size of water (~0.3 nm, molecular weight 18 g/mol) might compete with water molecules in penetrating into the hot spot. Even in such a situation where some small impurities are still left, however, we observe that the DDAA peak is not much changed in its position and bandwidth independently of the number of purification processes (refer to the spectra in the purification step 2 to 4 in Fig. 1d), which indicates that the small impurities do not contribute significantly to our SERS results. Moreover, more bulk organic species such as KCl, BH⁴⁻, glycerol, PVP and ascorbic acid are even more likely to be excluded from the hot spot, as indicated by the almost invariant DDAA peak shape (see the notes in Section 11 and Fig. S14 in Supplementary Information).”

Comment, Response and Revisions made 1-4.

The dilution factor from stage 1 to stage 5 with thirty-fold dilution is 8E5, not over 1E6.

We are sorry for the confusion. After three times dilutions from the beginning to the first stage, the solutions were diluted 30 times at each step so that the total dilution factor was $3 \times 30 \times 30 \times 30 = 2.43 \times 10^6$.

In page 4, we made clear in the revised manuscript by adding the following phrase:

“(i.e., the total dilution factor is 2.43×10^6)”

Comment, Response and Revisions made 1-5.

There is a small high-energy shoulder in the stage 5-purified SERS spectra of water, which could correspond to a very low fraction of free OH bonds. What fraction of free OH bonds are fitted if a 3-peak model fit is used for nanoconfined as well as bulk water?

In compliance to the reviewer's comment, we fitted both the bulk and the nanomeniscus water peak with 3-peak model and found that the fraction of the free OH bands are 8.9 % and 7.8 %, respectively. Accordingly, we revised Fig. 1d as well as the related discussions in the main text as follows.

In page 5, we added the following sentence.

“Moreover, the slightly lowered fraction of free-OH peak in the SERS signal (from 8.9 % for bulk Raman to 7.8 % for SERS, obtained by comparing the respective green-shaded area) suggests that water molecules are more hydrogen-bonded by tight confinement (DDAA or DA) and/or partly adsorbed on silver (DA) than bulk water^{26,27}.”

Comment, Response and Revisions made 1-6.

Clarify in figure caption of Fig. 2g that the ice spectra are taken from literature, by e.g. changing “all ice phases shows” on line 424 to “all ice phases (see Fig. S2 for their references) shows”. Furthermore, are all spectra performed with normal (bulk-sensitive) Raman spectroscopy or SERS?

We modified the figure caption of Fig. 2g as the reviewer suggested. And, at the end of the figure caption, we additionally mentioned the technique used by adding a sentence as follows:

“Note that all the Raman spectra reported in the references were performed by normal Raman spectroscopy, not by SERS.”

Comment, Response and Revisions made 1-7.

Please indicate the estimated internal pressure of ~1.37 GPa of your blue data point measured at 300 K in Fig. 2g.

Complying with the comment, we added the indication of the pressure in the revised Fig. 2g.

Comment, Response and Revisions made 1-8.

Please change “supercompressed-ice-VII” to “ice VII from supercompressed water” at 300 K,

since it is not (according to ref. 16 of the SI) the ice VII phase that is supercompressed but the liquid water phase from which it is transformed.

Complying with the comment, we changed the corresponding expression in the manuscript, the caption of Fig. 2. We also changed Fig. 2 itself that addresses the reviewer's comment.

Comment 1-9.

How does the peak at $\sim 200\text{ cm}^{-1}$ presented in Fig. 2c look like at drying stages 1-4 and for the various nanoparticle systems presented in Fig. 3c-d? Since I suspect the lattice vibrations would be mediated by the lattice vibrations in the nanoparticles, it would be highly interesting to see whether the peak changes for different systems. Furthermore, could the peak be assigned to nanoparticles or impurities solely, or does it only occur for nanoconfined water in between nanoparticles?

Response 1-9.

Thank you for the valuable comments. The following additional figure presents the low-frequency bands that behave in correlation with the OH stretching bands in various systems. In an answer to the first comment raised by the reviewer, we already showed that the peak at $\sim 200\text{ cm}^{-1}$ is not associated with the lattice vibration of silver metal. The following figure once again shows that both in the washing stages 1-3 (a and b) and in the various kinds of nanoparticles (c and d), the low-frequency bands show the correlated change with respect to the OH stretching bands only in the region of $150\text{ cm}^{-1} \sim 300\text{ cm}^{-1}$, without any indication of a significant change that could be induced by the change of lattice vibration of metal. Therefore, we observe again that the low-frequency peak is more likely to be assigned to the nanoconfined water, not to the nanoparticle metal or organic impurities.

Revisions made 1-9.

We added the following figure, Fig. S13, in Supplementary Information.

We also added the following sentence in page 12;

“More experimental details on the fabrication methods and characterization results as well as the corresponding SERS spectra including the low-frequency spectra are presented in Supplementary Information (see Fig. S2 and Figs. S10 to S13).”

Figure S13. Correlated behavior between low and high frequency bands observed in the washing stages (**a** and **b**) and in the various kinds of nanoparticles (**c** and **d**).

Comment 1-10.

There will be a strain on the synthesized core-shell nanoparticles so that the surface metal atoms will conform to the underlying lattice of the Ag shell. Thus, although it alters the chemical activity of the surface, it does not correspond to a pure nanoparticle of the shell element, see e.g. enhanced catalytic activity of Pt@Ru in Alayoglu et al., [Nature Mater. 7, 333 (2008)].

Response 1-10.

We agree with the reviewer's opinion in that there will be a strain so that the surface metal atoms will conform to the underlying metal lattice (Ag). Nevertheless, our system is different because the gold or platinum shell has at least more than 3 nm thickness, rather than 1-2 monolayer-thickness shell of atoms as the reviewer mentioned (please see Figs. S7d and S7h that present the EDS elemental line scan). Moreover, the highly decreased SERS activity in Pt@Ag structure also supports the existence of thicker shell of platinum, which contributes to the poor plasmonic optical activity (i.e., signal enhancement in SERS) compared to gold or silver.

On the other hand, even if the shell atoms conform to the underlying lattice of core atoms (Ag) in our core-shell structures, it is clear that the character of shell atoms in the core-shell particles cannot be the same with that in pure nanoparticle of the core element (Ag) (see the apparent differences in NC peak position that depends on the surface type as shown in Fig. S8). Notice that the purpose of our experiments on the examination of various nanoparticles is to demonstrate the generality of the molecular structure of the water nanomeniscus. Therefore, if one can observe the same spectrum (DDAA) of the nanomeniscus SERS even for slightly different surfaces, it can be the evidence to support our claim that there exists the generality of molecular structure of the water nanomeniscus.

However, complying with the reviewer's comments, we would like to mention in more detail on the conformation issue that can be present in the core-shell nanoparticles in our revised manuscript as follows.

Revisions made 1-10.

In page 11, we added the following sentences at the end of the paragraph.

“Note that there might exist a strain on the synthesized core-shell nanoparticles so that the surface metal atoms will conform to the underlying lattice of the Ag shell.⁴⁷ However, our gold or platinum shell has more than 3 nm thickness, much beyond the 1-2 monolayer-thickness shell of atoms (refer to the EDS elemental line scan in Figs. S7d and S7h). And the highly decreased SERS activity in Pt@Ag structure also supports the existence of thicker shell of platinum, which contributes to the poor plasmonic optical activity (i.e., signal enhancement in SERS) compared to gold or silver. Therefore, the Au@Ag or Pt@Ag nanoparticle used in our study exhibits the pure gold- or platinum-like surface characteristics, which is manifested by the different NC peak shift in Fig. S8.”

LANGUAGE

The manuscript conforms to the Nature Communications guidelines, but the language needs improvement. See specific suggestions below:

- ice phases are usually written with dash, e.g. “ice VII”
- add space between temperature and K, e.g. “300 K”
- p. 2, line 25: “carbon nanotube” → “carbon nanotubes”
- p. 2, line 27: “the wet sand” → “wet sand”
- p. 3, line 48: “anlaytes” → “analytes”

- p. 4, line 79: “is slightly observable” → “is only slightly observable”
- p. 5, line 87: “Obviously, the typical ...” → “The typical ...”
- p. 7, line 154: “.” → “.”
- p. 8, line 193: “(Au@Au)” → “(Au@Ag)”
- p. 10, line 229: “similar structure to the specific ice-VII” → “similar structure to specifically ice VII”
- p. 10, line 231: “can cover the stability fields at room temperature” → “are stable at room temperature”
- p. 22, line 239: “in the nanoscale water” → “in nanoscale water”
- p. 20, line 403: “of the bulk liquid water” → “of bulk liquid water”
- p. 20, line 405: “the smooth features” → “the broad features”
- p. 20, line 412: “implies the highly distorted” → “implies a distorted”
- p. 20, line 421: “at extremely low temperature” → “at low temperature”
- p. 20, line 426: “supercompressed-ice-VII” → “superexpanded ice VII” or “supercompressed water”
- p. 21, line 429: “ice-VII structure” → “ice-VII molecular structure”
- p. 21, line 441: “gold coated silver” and “platinum coated silver” → “gold-coated silver” and “platinum-coated silver”
- p. 21, line 442: “OH-SAM (2-mercaptoethanol) coated silver” → “OH-SAM (2-mercaptoethanol)-coated silver”
- p. 21, line 446: “its various size” → “its size”

Revisions made:

We appreciate very much for the detailed comments. As the reviewer pointed out, we have made complete corrections in the manuscript.

Reviewer #3 (Remarks to the Author):

Dear Editor

I have read the manuscript titled “Ultrasensitive Identification of Molecular Structure of Ambient Water Nanomeniscus” by Shin and coworkers.

Comment 3-1.

In their work, the authors report on surface-enhanced Raman spectroscopy study of water confined in inter-nanoparticle gaps. Based on the observed Raman spectra, the authors make conclusion that the water nanomeniscus has the stable molecular structure of ice-VII in ambient condition. I think this conclusion is absolutely wrong. The observed peaks in Raman spectrum (Fig. 1d) originates from the interfacial water, or reflects the PREWETTING transition, i.e. formation of a liquid layer at a surface. This phenomenon has been intensively studied for water in a slit pore of width of 24 Å (J. Phys. Chem. C 2007, v. 111, 15716), which is about the same size of inter-nanoparticle gaps. The peculiarity of water is the formation of water bilayer (water monolayer) upon prewetting transition. This water bilayer (monolayer layer), together with interfacial water is known as “bound water”, which show strong

positional and orientational ordering (ice-like). Thus this manuscript is not deserving publishing in Nat. Comm.

Response 3-1.

Thank you for valuable comments.

First of all, we would like to emphasize that the main claim of our work is not just to observe the “presence” of the interfacial (prewetting) water layer, but to further elucidate its “molecular structure” experimentally, which is found to be very similar to the structure of ice-VII. No matter how our inter-particle nanoconfined water is called, the interfacial water (prewetting) or the hydration water or the mixture of both, the important point of our work is that such nanoscale-confined water exhibits the ice-VII-like molecular structure. Notice that in our work, the water nanomeniscus is considered to consist of two components; the surface water layer (or strongly bound water according to the reviewer’s opinion) and the nanoconfined water (or weakly bound water), which are manifested by the DA and DDAA peaks, respectively.

Secondly, we certainly agree with the reviewer in that there have been extensive theoretical studies regarding the interfacial water, but it has been very challenging to demonstrate experimental clarification of the molecular structure in ambient conditions. Only a few works on the molecular structure have been reported until now, especially by using the nonlinear optical techniques (sum-frequency generation) or by using the artificial confinement structures (graphene or carbon nanotube). Notice that our work presents a novelty in that we have used a very natural confinement system of dried silver nanoparticulate in ambient conditions and succeeded in demonstrating its molecular structure by a common analyzing technique (surface-enhanced Raman spectroscopy).

In this regards, our work should be valuable enough to be considered as an important contribution to fundamental understanding of water behavior in the highly nanoconfined system.

Revisions made 3-1.

In compliance with the reviewer’s comment, we have modified in the manuscript to emphasize the importance of our work. In page 2, we added the following sentences at the end of the first paragraph.

“And, even though there have been extensive theoretical studies regarding hydration (interfacial) water, experimental clarification of its molecular structure in ambient conditions has been very challenging. In this article, by using the natural confinement system of dried silver nanoparticulate we have succeeded in demonstrating its molecular structure by a common analyzing technique of Raman spectroscopy, showing that the water nanomeniscus has the similar hydrogen-bonding character of ice-VII even in ambient conditions.”

In the middle of 2nd paragraph in page 5, we also added the following sentences:

“Notice that the water nanomeniscus can be considered to consist of two components; the surface water layer (or strongly bound water) and the nanoconfined water (or weakly bound water), which are manifested by the DA and DDAA peaks, respectively.”

Comment 3-2

In what follows, I will give few reasons to support my choice.

According to the water phase diagram, Ice VII is formed from liquid water above 3GPa (very high pressure). However, without applying pressure in the experiments, the claimed conclusion that the water nanomeniscus has the stable molecular structure of ice-VII is unconvincing.

Response 3-2

First of all, we would like to mention that the minimum pressure in the phase field diagram of ice-VII is 2.1 GPa, not above 3 GPa. Moreover, concerning the pressure generated in confined space, the similar nanoconfinement-induced high pressure effects in ambient conditions have been already reported in other previous experiments and theory:

- (i) Experimentally, the KI (potassium iodide) nanocrystal confined in carbon nanohorns (2 nm diameter) showed the super-high-pressure B2-phase structure (bcc), which occurs in the bulk KI crystals above 1.9 GPa.

K. Urita *et al.* JACS **133**, 10344 (2011).

- (ii) The conducting sulphur phase which can be only observable at the bulk scale above ~90 GPa was also obtained inside carbon nanotube in ambient conditions.

T. Fujimori *et al.* Nat. Commun. **4**, 2161 (2013)

- (iii) Theoretically, molecular simulation on the pressure tensor of argon molecules confined in nanoslit showed the highly enhanced pressure near the slit surface.

Y. Long *et al.* J. Phys. Chem. **139**, 144701 (2013)

Secondly, with regard to our experiment, the nanospace available in the purified silver aggregates and the fast timescale of ambient-drying process can render a supercompression-like effect on the system, exerting high pressure on the nanomeniscus. Simple estimate calculated by the interaction of water molecules with image dipoles on the silver surface also shows the pressure exerted is on the order of ~1 GPa (Supplementary Note 1), being consistent with the extrapolated pressure expected at the measured peak of 3340 cm⁻¹ (Fig. 3g).

All these results support that our nanoparticulate system may provide the pressure high enough to produce the ice-VII-like structure.

Revisions made 3-2

Considering the reviewer's comment, we added the above discussion including the references in the revised manuscript. In page 8, we added the following paragraph:

“In addition, we would like to note that similar nanoconfinement-induced supercompression effects in ambient condition have been reported in other experiments and theory: (i) Experimentally, the KI (potassium iodide) nanocrystal confined in carbon nanohorns (2 nm diameter) showed the super-high-pressure B2-phase structure (bcc), which occurs in bulk KI crystals above 1.9 GPa.³⁸ The conducting sulphur phase was also obtained inside carbon nanotube, observable at bulk scale above ~90 GPa.³⁹ (ii) Theoretically, molecular simulation on the pressure tensor of argon molecules confined in nanoslit showed the highly enhanced

pressure near the slit surface.⁴⁰ In our experiment, the nanospace available in the purified silver nanoaggregates and the fast timescale of ambient-drying process may render a supercompression-like effect on the system, exerting high pressure on the nanomeniscus. Simple estimate also shows the pressure exerted is on the order of ~ 1 GPa (details in Supplementary information), consistent with the extrapolated pressure expected at the measured peak of 3340 cm^{-1} (Fig. 2g).”

Comment 3-3

Moreover, the highest frequency available for ice-VII is 30 cm^{-1} lower than the obtained SERS of water nanomeniscus (Fig. 2g) can further support that the molecular structure of water nanomeniscus is different with that of ice-VII, although the authors tried to explain the difference.

Response 3-3

Although we mentioned only briefly in the manuscript, there have been sufficient precedent researches on the supercompression-induced abnormal phase change.

In the very early work by Yamamoto, it was discovered that ice-VII can be found in the metastable state within the stable region of ice-VI, as confirmed by X-ray diffraction precession photography. K. Yamamoto *Jpn. J. Appl. Phys.* **19**, 1841, (1980).

In particular, in more recent work by Lee *et al.* (ref. 37 in the manuscript), they reported the observation of ice-VII, directly crystallized from the metastably supercompressed liquid water at 1.72 GPa, well within the stability field of ice-VI. This was achieved by making time-resolved measurements of pressure-induced crystallization using a dynamic diamond anvil cell (dDAC), which allowed the measurement of the pressure/time-dependent phase transformation pathways. They were able to evaluate the interfacial free energy and find that the value for the supercompressed water (SW)/ice-VII is smaller than that of the SW/ice-VI, indicating that the local order of SW is more similar to ice-VII than ice-VI. Moreover, their result is consistent with the recent studies, which suggest that the local order of high density water is bcc-like, as in the ice-VII case. Notice that their supercompressed ice-VII exhibits the Raman peak position at 3335.5 cm^{-1} , almost 30 cm^{-1} lower than the one observed in normal ice-VII.

Remarkably, our nanomeniscus water shows the peak position at 3340 cm^{-1} , very similar to that of the supercompressed ice-VII. Moreover, considering both the fast timescale of ambient-drying process and the estimated pressure value exerted on water molecules in our system (see also our Response 3-2), we expect that our dried nanoparticulate system can provide a very similar pathway for the production of the supercompression ice-VII at room temperature.

Revisions made 3-3

In compliance with the reviewer’s comments, we have included the paragraph in page 8 in the revised manuscript as follows:

“Recently, a further increase of the DDAA frequency was reported at a lower pressure in the ice-VII that was directly crystallized from supercompressed water.^{37,38} This was achieved by making time-resolved measurements of pressure-induced crystallization using a dynamic diamond anvil cell (dDAC), which allows the measurement of the pressure/time-dependent

phase transformation pathways. The authors were able to evaluate the interfacial free energy and find that the value for the supercompressed water (SW)/ice-VII is smaller than that of the SW/ice-VI, indicating that the local order of SW is more similar to ice-VII than ice-VI. Moreover, their result is consistent with the recent studies, which suggest that the local order of high density water is bcc (body-centered-cubic)-like, as in ice-VII (Fig. 2f). Their ice-VII from supercompressed water exhibits Raman spectrum (green curve, Fig. 2e) which has its peak position at 3335.5 cm^{-1} (green square, Fig. 2g) even at 1.72 GPa, almost 30 cm^{-1} lower than the one in the normal ice-VII (marked by a short red-dotted arrow). Remarkably, our nanomeniscus water shows the peak position at 3340 cm^{-1} , very similar to that of ice-VII from supercompressed, which has a dense structure.”

Comment 3-4

Ice-VII has a density of about 1.65 g/cm^3 , which is much higher than the density of bulk water. As we known, the water layers next to the hydrophilic surface is very dense, and maybe that's why observed SERS of water nanomeniscus is like that of ice-VII.

Response 3-4

Once again, we would like to emphasize that the main claim of our work is to further “elucidate” the molecular structure, not just to observe the “presence” of interfacial (prewetting) water layer (please refer to Response 3-1). As the reviewer noted, it is well known that ice-VII has a dense molecular structure so that its hydrogen bonding network (tetrahedral) should be distorted (weakened) as well. In our work, we have succeeded in observing the Raman spectrum of this dense water (water nanomeniscus) and assigned it to the very similar peak position of ice-VII.

Revisions made 3-4

In compliance with the reviewer’s comment, we mentioned about the dense water layer in the revised manuscript. At the end of 2nd paragraph in page 8, we have already included the following sentences which are identical to the latter part of Revisions made 3-3.

“Moreover, their result is consistent with the recent studies, which suggest that the local order of high density water is bcc (body-centered-cubic)-like, as in ice-VII (Fig. 2f). Their ice-VII from supercompressed water exhibits Raman spectrum (green curve, Fig. 2e) which has its peak position at 3335.5 cm^{-1} (green square, Fig. 2g) even at 1.72 GPa, almost 30 cm^{-1} lower than the one in the normal ice-VII (marked by a short red-dotted arrow). Remarkably, our nanomeniscus water shows the peak position at 3340 cm^{-1} , very similar to that of ice-VII from supercompressed, which has a dense structure.”

Comment 3-5

After heating at 393K, the DA component remains, which indicates that the bound water molecules on the silver surface is ice-like (Fig. 2c). The disappearance of the DDAA component maybe related to the second layering transition or the disappearance of the second water layer near surface.

Response 3-5

Again, we agree with the reviewer's point that the selective decrease of the DDAA peak is related to the second water layer, which may be also called as the weakly bound water or the nanoconfined water (please refer to our Response 3-1). We emphasize that the importance and novelty of our work lies in the fact that the molecular structure of this second water layer exhibits the structural behavior very similar to ice-VII. The selective decrease phenomenon that the reviewer mentioned is another interesting observation that have we made, which is also well addressed in the manuscript (page 6 in the main text and Fig. 2).

Reviewer #4 (Remarks to the Author):

In this paper Shin et al use Raman spectroscopy to identify the dominating form of molecular organisations in water nano-menisci between metal nanoparticles at room temperature. They find unambiguously the water to adopt the molecular structure of ice-VII.

The results are in my view remarkable because ice-VII is normally expected only for water under extreme conditions of high pressure or low temperature. The existence of water in ice-VII form at room temperature opens up questions regarding our understanding of water behaviour at nano-interfaces. This has potentially some broad implications given the ubiquity of water nano-menisci in nature and technology.

However, while I find the results in principle exciting, I have some serious concerns with the paper in its current form.

Comment 4-1.

First, the paper almost exclusively focus on control experiments and misses the in my view the most important part: a thorough discussion about the implications of the findings (beyond the usual sweeping generalities). Do the findings have implications for the dynamics of water molecules in nano-menisci, and hence any chemical reaction that could take place? Are the results expected to stand in the presence of solutes? Do the findings have implications for the current use of Raman spectroscopy to identify water in extreme environments? Currently the paper is very dry and somehow very defensive, constantly trying to argue against possible objections. This makes the read off putting for non-Raman specialists (broad readership).

Response 4-1.

We really appreciate the reviewer's valuable comments. We have modified the manuscript to include discussions on the implications of our findings.

Revisions made 4-1

We added the following sentences at the end of the introduction paragraph in the revised manuscript.

“The results imply that, for example, the chemical reaction taking place in the water nanomeniscus should behave very differently with respect to that in the conventional bulk water or ice due to the substantially-weakened hydrogen bonding configuration of nanoconfined water molecules. Moreover, from a technical point of view, our results pave the new and efficient way of using the SERS on the trace amount of water in the extreme (nanoconfined) environment.”

Comment 4-2

Second, I am not convinced about the generality of the findings as claimed in the abstract. It is true that the authors make every effort to probe interfaces beyond noble metals, but even the use of SAM-coated Ag NPs is in my view not enough to assert that water nano-menisci between the grains of a drying sand-castle are indeed ice-VII. Besides, the authors go to great length to purify their water beyond anything expected for a general/natural system.

Response 4-2

Thank you very much for valuable comments.

We agree with the reviewer's opinion that our purification process seems to make the system further away from the real situation since there surely exist various impurities therein.

However, we would like to emphasize that the reason why we use purification process is to realize the clear optical detection of the trace amount of water by capturing it in the SERS hot spot, which provides highly nanoconfined space. Please note that even in the situation where some impurities are still left, it is likely that the DDAA peak is not significantly changed in its spectral position and bandwidth (see the spectrum in the purification step 2 to 4 in Fig. 1d in the main text). Moreover, we can observe the almost invariant DDAA peak shape even in the situation that various kinds of impurities are included, such as KCl, BH_4^- , glycerol, PVP and ascorbic acid (see the notes in section 11 and Fig. S13 in Supplementary Information). The results show that the impurities do not much affect the spectral shape of SERS spectrum of the water nanomeniscus, as far as water is concerned for the SERS response.

In order to realize the SERS detection of the water nanomeniscus, especially in the natural sands, we consider that the most probable way is to use the SHINER particle, which can provide both the optical Ag core and the chemical silica surface. However, as already noted in the section 11 in Supplementary Information, we have found that the very small thickness (over 1 nm) of the SAM on the silver nanoparticle makes it difficult to observe the SERS of the water nanomeniscus. Nonetheless, we have tried most of thin SAM coating that has the OH group, and succeeded in observing the SERS of water nanomeniscus on the surface very similar to the sand surface.

Revisions made 4-2

In compliance with the reviewer's comments, we have included the paragraph at the end of the paragraph in page 4 in the revised manuscript as follows:

“Even in such a situation where some small impurities are still left, however, we observe that the DDAA peak is not much changed in its position and bandwidth independently of the number of purification processes (refer to the spectra in the purification step 2 to 4 in Fig. 1d), which indicates that the small impurities do not contribute significantly to our SERS results. Moreover, more bulk organic species such as KCl, BH^+ , glycerol, PVP and ascorbic acid are even more likely to be excluded from the hot spot region, as indicated by the almost invariant DDAA peak shape (see the notes in Section 11 and Fig. S14 in Supplementary Information).”

Question 4-3

Finally and in the same vein, perhaps I am missing something, but the NPs seem citrate-coated to me given the protocol used to create them. I am not specialist on Ag NPs, but Au NPs will spontaneously decompose without a protective layer of citrate or a SAM. To me this raises serious questions about the picture given in Figs 1-2: where is the citrate? And what is actually being measured in terms of adsorbed water? I don't necessarily think that this invalidates the findings/conclusion, but it at least deserves a proper discussion, especially since the authors contrast 'Ag NPs' with 'SAM-Ag NPs', the later exhibiting a signal considerably poorer.

Response 4-3

We appreciate for very valuable comments.

We have created the Ag NP by the citrate reduction method, one of the most popular way of Ag NP synthesis. During the purification process (repeated centrifugation and dispersion step), the aggregation phenomenon is easily observed (even with naked eyes) due to the removal of surface citrate which is used to stabilize the nanoparticle as a suspension state. However, even in this situation, it does not mean that the nanoparticle spontaneously decomposes; rather it loses its well-dispersed state. Thus, the multiple purification steps are very critical in our success of achieving the SERS spectrum of the water nanomeniscus. Such a purification process removes, in particular, the surface citrate species, including various other impurities. Therefore, the scheme of Fig. 1a shows the highly purified state of nanoparticle aggregates, already being removed of the surface citrate species.

Revisions made 4-3

In compliance with the reviewer's comments, we have modified the manuscript as follows.

We added the following sentence in the beginning of page 4;

“Notice that during such a purification process, the aggregation phenomenon is easily observed (even with naked eyes) due to the removal of the surface citrate which is used to stabilize the nanoparticle as a suspension state.”

And in the caption of Fig. 1, we modified as in the following sentence;

“a, Schematic showing the trapped water nanomeniscus in the optical hot-spot region after the highly purified silver nanoparticles (with the surface capped-citrate removed) are dried in ambient conditions.”

Reviewer #5 (Remarks to the Author):

This is a very interesting paper that should be published. The use of SERS in conjunction with water capillaries is creative and reveals a rather unique spectral response of confined water. I have a few comments the authors may wish to consider.

Comment 5-1.

For the sake of scientific rigor, I would suggest the authors to tone down the claim that the water is ice; it has the same vibrational response of ice, but that does not mean that the structural properties, like $g(r)$, are the same. It would not hurt, and improve the paper, I submit.

Response and Revisions made 5-1.

Thank you very much for kind comments and we fully agree with the reviewer.

In compliance with the reviewer's suggestion, we have made a tone-down correction in our claim of ice-VII in the entire manuscript by changing the expression “ice-VII” to “ice-VII-like” or “similar molecular character of ice-VII”. Please see the blue-colored words in the Abstract, page 2, page 10, page 12, Fig. 2 caption and Fig. 3 caption.

We also added the following sentence at the end of the first paragraph in page 9; “We would like to emphasize, nonetheless, that although the SERS spectrum exhibits the same vibrational response of ice-VII, it does not necessarily mean that the structural properties, such as the molecular density distribution, are the same, which has to be further investigated.”

Comment 5-2.

The authors should do at least one set of experiments with isotopically diluted H₂O/D₂O mixtures. It is well-known (see e.g. papers from Skinner, Bakker, and others) that the OH stretch vibration is delocalized, especially in ice. This excitonic behaviour depends on the size of the water cluster under study. Hence, the most appropriate comparison would be to study the O-H stretch spectrum of isotopically diluted HDO in D₂O. The point is that it is not clear how many water molecules are in the meniscus relative to the delocalization length of the OH stretch.

Response 5-2.

According to the reviewer's suggestion, we have performed an isotopic experiment to investigate the OH stretching spectrum, but unfortunately, based on the following control

experiment we could not stably observe the OH stretching bands of isotopically diluted HDO in D₂O in our system.

At first, we tried to synthesize silver nanoparticle in the environment of all D₂O condition. All precursor chemicals (silver nitrate and sodium citrate) were prepared as dissolved in D₂O and boiled in D₂O solvent as well. Produced silver nanoparticles were also purified with D₂O solvent and dried in ambient condition.

However, we interestingly have observed, as shown in the following Fig. R1, that the OD stretching band (near 2500 cm⁻¹) becomes totally disappeared during drying process while only the OH stretching band (near 3500 cm⁻¹) is left. Moreover, during drying, the OH stretching band is accompanied by the slight shift of the peak maximum of the low-frequency band near 200 cm⁻¹ (i.e., the lattice vibration bands of ice). These results imply that there is an exchange phenomenon between the highly-confined D₂O and the ambient H₂O, occurring even in the highly confined space of trapped region within the dried nanoparticulates.

Therefore, even though we cannot stably observe the OH stretching spectrum of isotopically diluted HDO in D₂O in our system, we observe another interesting exchanging behavior (or mass transport) in the ultraconfined space. We hope to examine this unexpected phenomenon further in detail as a future work.

Figure R1. Drying process of all D₂O-based silver nanoparticle solution in ambient condition exhibits unusual SERS spectrum change, assuming the occurrence of molecular exchange phenomenon at ultraconfined space.

Concerning the number of molecules in the nanomeniscus, we can roughly calculate by using the result from the FEM simulation method (real geometric value was derived from TEM observation) since most of SERS signal (over 99%) comes from the hot spot, which is highly

confined. We then obtain the number of ~775 molecules (at maximum) in a single hot spot region (please see the Supplementary Notes 1).

Revisions made 5-2.

Although we have conducted isotopic experiments and obtained results, we would rather not present the data in the present manuscript because one needs further work to make a definite claim. However, we added the following sentence at the end of conclusion in page 13;

“Notice that although we have performed a separate experiment of isotopically diluted HDO in D₂O, we could not stably observe the OH stretching spectrum, which is worthy of further systematic investigation.”

Comment 5-3.

The authors are advised to explore the role of the meniscus on the spectra: there will be a substantial LaPlace pressure, that may contribute to the high-pressure-type of behavior observed. A good starting point may be the book 'Intermolecular Forces' by Israelachvili.

Response 5-3.

Thank you for the critical and useful comment.

Indeed, the high surface curvature of the nanomeniscus is known to induce substantial Laplace pressure as the reviewer mentioned. On the other hand, as also noted in the book of Israelachvili, it should be mentioned that the Laplace pressure always drives the interface in the concave direction. Thus, the liquid in a droplet (convex surface) experiences a positive (compressive) pressure, whereas a liquid containing a bubble or hole (concave surface) experiences a negative (tensile) pressure. The positive pressure on a droplet means that its molecules are compressed, while around a concave surface the molecules are expanded.

Following such a line of explanation, our system of suspended water nanomeniscus between adjacent nanoparticles may have concave interfaces so that the molecules should experience highly negative pressure, resulting in expansion. However, this explanation seems to contradict with our observation of ice-like character of the nanomeniscus, which is spectroscopically evidenced by the existence of the lattice vibration band as well as the highly narrowed bandwidth of the OH stretching band, together with other various evidences supporting the existence of the high pressure ice phase. Moreover, our ultraconfined geometry formed between dried nanoparticulates (Fig. 1b) may not provide available space for the existence of such an expanded water structure, even in the case of negative pressure. Thus, we consider this may not provide plausible explanation for our findings.

Instead, we would like to explain our findings by the surface-induced high pressure, which may be more relevant to our results, associated with the nanoconfinement-induced high pressure effects in ambient condition. Such an effect already has been reported in other previous experiments and theory:

- (i) Experimentally, the KI (potassium iodide) nanocrystal confined in carbon nanohorns (2 nm diameter) showed the super-high-pressure B2-phase structure (bcc), which occurs in bulk KI crystals above 1.9 GPa.

K. Urita *et al.* JACS **133**, 10344, (2011).

- (ii) The conducting sulphur phase, which can be only observable at bulk scale above ~90 GPa, was also obtained inside carbon nanotube in ambient condition.

T. Fujimori *et al.* Nat. Commun. **4**, 2161, (2013)

- (iii) Theoretically, molecular simulation on the pressure tensor of argon molecules confined in nanoslit showed the highly enhanced pressure near the slit surface.

Z. Long *et al.* J. Phys. Chem. **139**, 144701, (2013)

Besides the Laplace pressure, we added related discussion in the revised manuscript as follows.

Revisions made 5-3.

In page 8, we added the following paragraph, which is the same as our Revisions made 3-2:

“In addition, we would like to note that similar nanoconfinement-induced supercompression effect in ambient condition has been reported in other previous experiments and theory: (i) Experimentally, the KI (potassium iodide) nanocrystal confined in carbon nanohorns (2 nm diameter) showed the super-high-pressure B2-phase structure (bcc), which occurs in bulk KI crystals above 1.9 GPa.³⁸ Moreover, the conducting sulphur phase was also obtained inside carbon nanotube, observable only at bulk scale above ~90 GPa.³⁹ (ii) Theoretically, molecular simulation on the pressure tensor of argon molecules confined in nanoslit showed the highly enhanced pressure near the slit surface.⁴⁰ In a similar way, the nanospace available in the purified silver nanoaggregates and the fast timescale of ambient-drying process in our experiment may render a supercompression-like effect, exerting high pressure on the nanomeniscus. Simple estimate also shows the pressure exerted is on the order of ~1 GPa (details in Supplementary Information), consistent with the extrapolated pressure expected at the measured peak of 3340 cm⁻¹ (Fig. 2g).”

REVIEWERS' COMMENTS:

Reviewer #1 (Remarks to the Author):

The authors clarified all my concerns in the revised version of the manuscript and the rebuttal letter.

I now recommend this manuscript for publication in Nature Communications.

Reviewer #3 (Remarks to the Author):

Both the Referee #5 and I thought the water is not the ice-VII, although the same vibrational response of ice-VII. The author then changed the expression "ice-VII" to "ice-VII-like" or "similar molecular character of ice-VII". I agreed the author that understanding the molecular structure of water under nano-confinement experimentally is very important. However, I would not suggest to publish this paper if there is no comparison on the structure of water between at the interface and under confinement, since I suspect observed peaks in Raman spectrum have nothing to do with nano-confinement.

Reviewer #4 (Remarks to the Author):

The authors have addressed my concerns and I am satisfied with the revised manuscript. I therefore recommend publication.

Reviewer #5 (Remarks to the Author):

The authors have carefully addressed my concerns and those of the other reviewers. The H/D exchange implies that there is ambient humidity during the 'drying' - but that should not prevent publication of this manuscript.

1. Response to the Reviewer #3's comment.

Reviewer #3 (Remarks to the Author):

Both the Referee #5 and I thought the water is not the ice-VII, although the same vibrational response of ice-VII. The author then changed the expression “ice-VII” to “ice-VII-like” or “similar molecular character of ice-VII”. I agreed the author that understanding the molecular structure of water under nano-confinement experimentally is very important. However, I would not suggest to publish this paper if there is no comparison on the structure of water between at the interface and under confinement, since I suspect observed peaks in Raman spectrum have nothing to do with nano-confinement.

Response:

We would appreciate very much for another but similarly repeated comment by Reviewer #3.

First of all, in addition to our original responses and revisions made with respect to several comments by Reviewer #3, we would like to emphasize again that the key idea of our work relies on the exact spatio-spectral overlap between the physically formed-nanomeniscus (i.e. capillary condensed water) region and the optically formed-hotspot (i.e. locally intensified electromagnetic field) region in our nanoaggregates system. Considering the fact that (i) water molecules should be thermodynamically capillary-condensed in the inter-nanoparticle gap (i.e. nanoconfined space) in ambient humidity condition and (ii) the SERS signal comes only from such nanoconfined hot-spot region, we can conclude that our water signal originates from the nanoconfined space, not from the interface.

Secondly, we would like to draw attention to another previously-reported work¹, where it is shown that the interfacial water (on the metal surface) has mostly in-plane hydrogen bonding geometry, which corresponds to the DA component in our Raman spectrum. Moreover, selective decrease of the DDAA component associated with heating, without any change of the DA one, strongly suggests that the DDAA and DA components correspond to the nanoconfined and interfacial water, respectively.

Reference.

[1] Kimmel *et al.* Crystalline Ice Growth on Pt(111): Observation of a Hydrophobic Water Monolayer, *Phys. Rev. Lett.*, **95**, 166102 (2005).